# Subjective Experiences of At-Risk Children Living in a Foster-Care Village Who Participated in an Open Studio

**DOI:** 10.3390/children9081218

**Published:** 2022-08-12

**Authors:** Michal Bat Or, Reut Zusman-Bloch

**Affiliations:** 1School of Creative Arts Therapies, University of Haifa, Haifa 3498838, Israel; 2Ministry of Education Israel, MATYA Zvulun Asher, Kiryat Ata 2804062, Israel

**Keywords:** at-risk children, open studio, art, foster care

## Abstract

The open studio art therapy model offers a space for free creation; in this space, the art therapist supports the participants’ art process. According to this model, the creative process is the central component of the therapeutic work. This qualitative study seeks to learn, through an analysis of interviews and artwork, about the subjective experiences of at-risk children living in a foster-care village who participated in an open studio. In addition, it seeks to identify changes in the artwork over time. This study involves a qualitative thematic analysis, while the analysis of visual data is based on the phenomenological approach to art therapy. The data include interviews and 82 artworks of five participants, aged 7–10 years. Five main themes emerged from the analysis of the visual and verbal data: (a) engaging in relationships; (b) moving along the continuum from basic, primary, art expressions (e.g., smearing, scribbling, etc.) to controlled expressions; (c) visibility, on a range between disclosure and concealment; (d) holding versus falling/instability; and (e) experiencing and expressions of change. The discussion expands on the themes in relation to key concepts in the field of psychodynamic psychotherapy and art therapy. It also examines the unique characteristics of this population in reference to empirical studies on developmental trauma and challenges of out-of-home placement. Finally, it discusses the study’s limitations and presents recommendations for further research.

## 1. Children at Risk in Foster Care

The phrase “children at risk” underscores the threat of future negative consequences of risk factors, such as anxiety, depression, dropout from school, addiction, or early death [1]. Prevalent at-risk situations include abuse and neglect; these can have far-reaching effects on children’s emotional, cognitive, and physical health [2]. Most at-risk children who are removed from their homes come from a background of serious family problems [3].

Out-of-home placements are interventions by the Ministry of Welfare designed to protect children from the risks of continuing to live with their harmful families. In Israel, 70.1% of out-of-home placements involve boarding schools (some in the form of foster-care villages) that provide a therapeutic and rehabilitative framework [4].

Studies have observed a high incidence of psychological, social, and educational problems among children in out-of-home placements in comparison to the general population, for example, a recent systematic review and meta-analysis of prevalence observed that suicide attempts were more than three times as likely in children and young adults that were in out-of-home care [5]. Health screenings of 122 young children in out-of-home care observed that 54% had behavioral/emotional problems (e.g., poor self-esteem, anxiety, grief, problematic interpersonal relationships, and depression), 33% had a speech delay, the hearing test for 30% was abnormal, and 28% failed their vision test [6]. Recent studies also underscored the higher prevalence of sleep problems as also related to mental health problems and crime among out-of-home care children [7] and young people who were in out-of-home care as children [8]. These findings may indicate that children who have been removed from their homes have already experienced trauma, and that removal from the family may create unstable circumstances [9]. Houzel [10] coined the phrase “family envelope”, which provides the experiences of belonging and connectedness, but also the permission to differentiate themselves and to form individual identities. He suggested that in cases of dysfunctional families, therapeutic services may provide a “widened envelope”, by joint efforts of therapeutic professionals who provide a sense of belongingness together with the permission for children to differentiate themselves [10].

## 2. Childhood Trauma

Trauma refers to the effects of exposure to potentially extremely stressful events [11]. The direct or indirect exposure to abuse can cause physical, mental, and emotional trauma [12]. Trauma may change children’s perceptions of themselves and alter their connection to their bodies and the world around them; it usually leaves them feeling anxious, vulnerable, lonely, helpless, and hopeless [12,13]. From a neurobiological point of view, experiences of maltreatment can impair the function of the prefrontal cortex (PFC) that controls limbic system responses via the PFC–amygdala–hippocampus network. Therefore, unregulated signals from the amygdala could cause high anxiety levels due to inadequate cognitive discrimination, ending in affect dysregulation [14]. In addition, since the right hemisphere is dominant during the first three years of life and processes information in the form of nonverbal signals, early traumatic sensory-motor experiences are stored in nonverbal form and are accessible through bodily expressions [15]. This neuroscientific knowledge of underlying brain functions can increase the effectiveness of art therapy for these children [16] and may explain why art expression as a therapeutic method for children at-risk has its advantages.

## 3. The Visual Art Medium as a Therapeutic Tool for Treating Children at Risk

Although expressive arts are a developmentally appropriate activity for children that enable their voice to be heard [17], and a high percentage of art therapists work with children, e.g., [18], art therapy research on traumatized children mainly constitutes case studies and qualitative research (see review [19]). Creative expression for traumatized children has been oberved to be an effective therapeutic tool [20,21] because children communicate their emotions and thoughts through nonverbal means [17,22,23]; moreover, through art expression, children gain greater knowledge and awareness of themselves, and develop their self-esteem as well as social skills [20]. Resiliency refers to the ability to adapt, despite experiencing deficiency and severe distress [24]. Art therapy may promote resiliency among clients who suffer from adverse life events [22,25]. Artwork may therefore allow clients to play an active role in their own personal therapeutic processes [26].

One of the main approaches in art therapy is the “art as therapy” approach, in which artwork is therapeutic in nature [27]. One model based on this approach is the open studio model [28].

## 4. The Open Studio Model

The open studio was developed to serve larger populations and communities in need, by making art-creative processes accessible [29]. A recently conducted systematic scoping review of the open studio model [28] revealed several core principles, including the central role of art, the experiential dimension of the creation of artwork, and the art therapist’s/facilitator’s role of holding a space to allow for a free creative process [28,30]. The open studio model seeks to offer a space that allows for the creative expression of the individual and the group [31,32]; it provides an egalitarian, nonhierarchical, open-minded, and community-like environment [30,33].

Studies that examined the impact of the open studio model on different populations found that studio work increased positivity, significantly strengthened self-esteem [34], and enabled the expression of personal and relational themes (e.g., [35]) and painful emotions through playfulness and pleasure [36]. The study of an open studio pilot project in Illinois, USA, involving more than 100 children aged 10–12 years found this model particularly suitable for at-risk children who grew up in a chaotic and violent environment [33]. However, according to the research, the open studio simultaneously brought participants face to face with painful feelings and experiences [33]. Although the unique setting provides an adaptable framework for populations in need of therapy, only few studies have been conducted on the effectiveness of the open studio model among children and adolescents [28].

The present qualitative study examined the experiences and artwork of at-risk children, aged 7 to 10 years, living in a foster-care village. The study had two purposes: first, to examine, as reflected in the central themes of their artworks and in their verbal descriptions, the subjective experiences of at-risk children who live in a foster-care village and who chose to create in an open studio. The second was to analyze the children’s artworks with the purpose of identifying changes over time. Based on these goals, the two main research questions were: (1) based on their interviews and their artworks, what were the subjective experiences of foster-care village at-risk children who participated in an open studio? (2) Can we identify changes in the artworks over time, and if so, what was the nature of these changes?

## 5. Method

### 5.1. The Research Approach

The present study used a thematic analysis within a qualitative methodology [37]. A thematic analysis allows for the interpretation of various aspects of the research topic [38]. The analysis of the artwork was based on the phenomenological approach to art therapy, which involved a process of meticulous scrutiny of the content and formal features of the artwork [39].

### 5.2. Participants

The sampling method involved a deliberate selection [40] of five at-risk children (four girls and one boy) from a foster-care village in Israel, where an open studio operated three afternoons a week. The children regularly attended the open studio and their artwork was kept on the premises. A total of about 90 children attended the open studio over the year; the size of the groups ranged from five to fifteen children in a session of 90 min. The study participants’ ages ranged from 7 to 10 years (mean age: 8.4 years). The number of artworks created by each child ranged from 21 to 109, while the average number was 53. Table 1 presents the children’s age, gender, the number of times they attended the open studio during the year, and the number of artworks they created.

### 5.3. Research Tools

*Observation Rating Scales Sheet for Art Therapy Practice: Sections B1 and B2: Observation and analysis of drawings and three-dimensional artworks—sculpture/structure/textile [41]*.

This tool was used for art therapy education and practice; the rater is required to answer yes/no to a detailed list of formal and content features. The tool contained five main formal aspects: overall impression (for example, artwork with repetitions); line quality (for example, thick or weak line, and the presence of an eraser); color qualities (for example, and separated versus mixed colors); space (for example, the amount of space used and the presence of a frame); and finally, forms (for example, the presence of geometrical forms). In terms of content, the rater was asked to rate the artwork for its realism/abstraction and symbolism.

*Analysis according to the Expressive Therapies Continuum model (ETC)—The Expressive Therapies Continuum [42,43]*.

We analyzed the artworks according the ETC hierarchical model, ranging from simple kinesthetic expressions to complex symbolic images. Specifically, there were three levels of artistic expressions, with each level being arranged in a continuum between two poles. Each level may demonstrate a different way to process information [44]. The first and basic level ranged from kinesthetic components (for instance, evidence of smearing or pressing on the material) to sensory components (for instance, evidence of direct touches on the material, such as finger marks on clay); the second level ranged from perceptual components (for instance, evidence of differentiated forms and colors, contour-lines, etc.) to affective components (for instance, evidence of the expressive use of colors); the third level ranged from cognitive components (for instance, the addition of written words) to symbolic components (for instance, the expression of a metaphor, the combination of realism and abstraction, etc.); and the fourth level was the creative level and demonstrated the integration between levels/poles.

*FEATS: Formal Elements Art Therapy Scale [45]*. We used the verbal definitions of twelve 5-point Likert scales for drawings only: prominence of color; color fit; implied energy; space; integration; logic; realism; problem solving (only for drawings that depicted a person that must overcome an obstacle or achieve a goal); developmental stage; details of objects and environment; line quality; and person. Each scale depicted a range of options starting from a non-existent phenomenon (e.g., in the prominence of color scale, a drawing executed solely in pencil) and ending with the highest point, which could represent a drawing in which the whole space was filled with color. For abstract drawings, we only used relevant scales: prominence of color, implied energy; space; integration; and line quality.

Two art-therapy students (the second author was one of them) and an experienced art therapist and researcher (the first author) practiced the ratings of the three research tools for this study, and for an additional study with adolescents [35]. After the raters had reached an agreement in regard to the phenomenological aspects of the artworks during analysis, via careful observation, analysis, discussions, and continued integration of the relevant theoretical models, the second author continued the rating independently, and the second author participated in the reduction process of the integration into themes. The data codes were inserted into an Excel table for further data analysis and integration.

### 5.4. Procedure

After ethical approvals were obtained from the Israeli Ministry of Education and the Ethics Committee of the Faculty of Social Welfare and Health Sciences at the University of Haifa, we reached out to the education staff and social workers of the foster-care village and asked them to invite children who attended the open studio to participate in a study that focused on their experiences and artwork in the studio. About 15 children agreed; however, parental consent was given in only five cases. These children’s artworks were photographed for analytical purposes. The 255 artworks were classified into categories and a general phenomenological analysis was conducted for each category. Specifically, two-dimensional artworks were divided into three categories according to the materials from which they were made (liquid materials, controlled materials, and intermediates), whereas three-dimensional artworks were divided according to the following categories: structures; containers; figures; shapes; platforms surfaces; accessories; and a combination of categories. After this division, two to three artworks were selected from each category, and an in-depth analysis was conducted of 82 drawings using the three research tools. In addition, towards the end of the year, semi-structured, individual interviews were conducted with the children about their experiences in the open studio.

## 6. Data Analysis

The artworks were analyzed and implemented into a summary table based on the clinical observation sheets, the ETC model, and the FEATS rating system. The table addressed the following aspects in each artwork: a phenomenological description of the artwork; formal features; expression levels according to the ETC model; materials and methods used; content; the subjective experience of the viewer; and changes noted in relation to previous artworks. The analysis of the works of art yielded categories that were then incorporated into the six-steps thematic analysis [37].

The analysis process followed these steps: (a) familiarization with the data through reading the interviews and a close phenomenological observation of the artworks; (b) creating initial categories for data, while actively looking for meaningful patterns and relationships within the visual expressions as well as between the visual and verbal expressions; (c) sorting the categories into potential themes; (d) review of themes in the search for consistency in meaning as well as discrepancies; (e) definition, naming, and detailed analysis of each theme and subtheme, as well as a description of the relationships between different themes and the connection between the theme and research questions; and (f) writing a thematic analysis in the context of the theoretical model, including the findings, and the study’s validity and limitations.

Finally, for the purpose of examining the artworks for changes over time, the authors observed the summary table and the individual summary of each child’s artworks, and identified expressions of changes over time, including explicit verbal expressions in the children’s interviews and visual expressions in their artworks that showed a change in formal art elements and/or content representations.

## 7. Findings

Interviews with the children occurred at the end of the academic year. Two participants were cooperative and spoke at length about their subjective experience in the studio, while the other three participants presented brief answers. The interviews lasted between 10–30 min.

Five major themes emerged from the thematic analysis of the five interviews and a meticulous phenomenological analysis of 82 artworks of participants who attended the open studio over the year. These themes were (a) engaging in relationships; (b) moving along a continuum between basic primary art expressions (e.g., smearing, scribbling, etc.) and controlled expressions; (c) the need for visibility, on a range between disclosure and concealment; (d) holding versus falling/instability; and (e) experiencing and expressions of change. Bar Chart 1 illustrates the evidence of each of the four themes as they were detected in each child’s verbal and artwork expressions.

Bar Chart 1. About here.



### 7.1. Engaging in Relationships

This theme is the dominant theme, and was identified 57 times in the artworks and interviews (19 references in the interviews and 38 in the artworks, constituting 46% of the works analyzed in the study). The three subthemes were close relationships, loneliness, and relationships within the studio.

### 7.2. Close Relationships

There were 24 references to the subject of close relationships (7 artworks relating to the subject and 17 references in the interviews). Four out of five participants depicted a connection to a parental figure in their artwork. For example, the love for a mother was expressed through the use of transparent and scattered pieces of material (Figure 1: with the word “Mom” and a heart made of pieces of transparent and red materials). In some artworks, expressions of love were combined with aggression (e.g., artwork that contained nails), while in others, close relationships were related to pain and sadness (e.g., Figure 2: showing a crying figure). These artworks perhaps reflect the complex emotions felt towards family members.

In the interviews, children described their relationship with family members, for example, Keren said: “My brother has a double bed and he’s in the room alone with a TV, and a computer and all sorts of things […] And me and my sister are in the same room, it’s a nightmare, even sleeping in the same room with her, waking up at night because she’s taken my blanket […] I take my mother’s blanket and say thank you, goodbye”.

It may be assumed that Keren is also speaking implicitly about experiences of place and lack of place, as well as being protected or not protected by her caregivers.

### 7.3. Loneliness

Loneliness appeared only in the artworks (specifically, in 15 of the participants’ artworks) and not in the interviews; loneliness was expressed in several ways, such as single figures isolated from their environment, for instance, a lone figure on a blank sheet of paper (Figure 2 and Figure 3). Loneliness was also connected to misery and strong emotions (see Figure 2, which features the word “alone” written on the drawn tears).

In some artworks, loneliness was also expressed through the character’s relationship with the environment, e.g., a distinction and contrast between the figures and background (see Figure 4 showing a figure separated from the background or encapsulated).

Finally, in other artworks, the placement of figures emphasized the lack of communication between them (e.g., Figure 5 featuring many figures, but all facing different directions).

### 7.4. Relationships in the Open Studio

This subtheme was present in 18 references. Interviews revealed that all participants perceived the relationship with the therapist in the studio as positive, supportive, and enjoyable. When asked about the work experience with the therapists in the room, the participants (Keren, Dana, Alex, and Tamar) answered: “Fun”. In addition, the children described new and meaningful friendships they had formed during their work in the studio. The relationship with their peers seemed to create some challenge, as the participants spoke about the difficulties of creating within the group space. When asked if work in the open studio was frustrating, Dana answered: “The truth is, a lot of time, (laughing), there was one time, that someone, a girl took the piece I wanted to make, just took it from me without asking”. This and other examples demonstrated the frustration felt when personal space was invaded; moreover, all participants mentioned that the noisy environment bothered them.

Relationships within the studio were also reflected in joint artworks: four of the five participants in the study experienced this method of working (see Figure 1 and Figure 6).

Different children sometimes worked on portraying the same image (e.g., two children drew the YouTube logo or copied a penciled drawing of an Indian god). Working on the same image can also indicate imitation, mutual influence, reciprocity, and interpersonal connections within the studio.

Moving Along a Continuum Between Primary Art Expressions (e.g., Smearing, Scribbling, etc.) and Controlled Expressions.

This theme was expressed in the artworks of all the participants and specifically in 41 artworks, i.e., half of the total number of artworks analyzed in this study, whereas only two references to this theme were noted in the interviews.

## 8. Control

Regarding the materials, we examined artworks that demonstrated control and the meticulous use of liquid materials. Tamar, for example, stated her preferences: “Oil pastels, markers, colored pencils, papers,” but in contrast, her artworks were predominately created with liquid materials (see Figure 1 and Figure 4). Dana, on the other hand, showed a clear preference for controllable materials: 82% of her two-dimensional works were painted with materials of this kind, while her other artworks made with other materials also demonstrated control (see Figure 2 and Figure 3). There were also artworks that expressed the need for control through their content, for example, the depiction of figures with special powers (see Figure 4, showing a figure wearing a crown and wand, symbols of power and control).

### 8.1. Characteristics of Order and Recurrence

Aspects of organization and order, such as symmetrical images and repetitive movements or shapes, were found in 15 artworks (e.g., Figure 6). Organization was also represented by division into blocks of color, order of colors, and meticulous coloring, and symmetry was observed in artworks where there were two similar parts that demonstrated the same use of color or shapes (see Figure 6). Repetition was demonstrated through the repeated use of colors, shapes, and strokes (see Figure 7).

The need for control also appeared in the interviews: three out of five participants (Tamar, for example) stated their preference for controllable materials, but their artwork reflected the use of liquid materials that were less controllable.

### 8.2. Primary Art Expressions (e.g., Smearing, Scribbling, etc.) and Expressions of Chaos

The Kinesthetic/Sensory level—the first level of the ETC model that involves primal developmental expression—was reflected in 19 artworks by 4 participants (see Figure 8 that features a plastic container containing a crumpled page that was colored with gouache paint).

The prevalence of kinesthetic sensory artworks varied from one participant to another; some participants barely engaged with this level, while for others it comprised the main part of their body of artwork. The artworks dealt with the primary need for touch and movement [46] and may reflect the discovery of the ability to leave a mark [47]. These needs corresponded to the needs of the infant and toddler in the early stages of life; participants created pre-formal artworks, i.e., abstract artworks created from movement and contact with the material, which may represent a return to a primal state where form still has no meaning, such as syllables that have not yet formed a word. Tamar’s clay works show a transition from organization to disorganization, from form to lack of form, with the last work in the series being amorphous in nature (see Figure 9).

### 8.3. Visibility—Concealment and Discernibility

The theme of visibility was identified in 33% of all the artworks analyzed. Of these, 20 artworks directly presented the theme of visibility, while the other 33 works featured three additional aspects of visibility: secret (6), cover (15), and interior and exterior (12).

The analysis of the artworks revealed two main features of visibility. The first was the use of shiny materials, such as glitter, sequins, and golden copper (see Figure 10 and Figure 11). These visual materials may symbolize the need to exhibit something, attract attention, or stand out.

The second feature was apparent in the content that emphasized eyes, such as figures with highlighted eyes (Figure 12). In addition to emphasizing eyes, this theme was reflected in artwork that was made on or inside glass cases (Figure 10 contains these two features).

### 8.4. Secrecy

The theme of secrecy appeared a total of six times in the artwork of four participants. Secrecy was presented as part of the visibility theme and was reflected in the choices of concealment and disclosure. The artworks that dealt with this theme were all three-dimensional, characterized by closed containers with hidden contents. In her interview, Dana openly dealt with the theme of secrecy: “I just thought, let’s make a drawer: a drawer with all kinds of secrets inside” (Figure 13). By bringing the secrets into the studio in the form of artwork, the participants may attempt to share and work through their experiences with the staff and/or other children.

### 8.5. Cover

Another aspect of the theme of visibility is covering. Four participants addressed the issue of covering in 15 artworks. This is a topic that was only expressed in the artwork, mainly in material form, i.e., one material covering other materials (see Figure 11). The cover may reflect concealment in the sense that one chooses to cover and hide certain layers or parts of the artwork. It is possible that the very preoccupation with cover expresses a need for a protective layer, or alternatively, a desire to be discovered through the removal of these layers. It could also represent unmanifested or unconscious mental aspects.

### 8.6. Interior and Exterior

Connections or contrasts between the interior and exterior were identified in 12 artworks. In some of the artworks, a tension emerged between the interior hidden layers and the visible, exterior layers. Most of these artworks were containers that presented a contrast between the interior and exterior: the exterior reflected harmony and a high degree of control over material and color, while the interior reflected contrasts, i.e., a mix of colors and chaos (see Figure 14).

In conclusion, the theme of visibility is a dialectical one, and may reflect a tension between the desire for disclosure and visibility, and the wish to hide and conceal secrets, complex, uncontrollable, and primary elements.

### 8.7. Holding versus Falling/Instability

#### 8.7.1. Holding

By holding, we refer to the emotional experiences of being held by the other/the setting/artwork, as described by Ogden [48] following Winnicott [49,50]. The experience of being held beyond infancy is described as the provision of a “place” in which the client may gather himself/herself together, including all his/her different parts/aspects. Since metaphors of an envelope, a frame, or a container are often considered by therapists as describing some aspect of the holding function (e.g., [51]), it may be assumed that the multiple use of these elements in these artworks communicates the holding function, and/or the need to be held and integrated.

The holding theme was manifested in 18 artworks in two main ways. One was the presence of frames (Figure 2) and the other was the use of containers (Figure 14). The only frames that appeared to be complete in the artworks were ready-made frames.

Among the three-dimensional artworks, we found containers, such as boxes, purses, pencil cases, bags, bottles, and more. In many artworks, the container contained liquid/soft materials, such as clay, gouache, and even a combination of materials, including water (see Figure 8 and Figure 14). In addition, some of the containers were food containers and their selection may indicate the primal need for nourishment, also connected to maternal holding.

#### 8.7.2. Falling and Instability

Themes of falling, seepage, and instability were observed in ten artworks. Three-dimensional artworks reflected instability, while two-dimensional artworks depicted situations in which there was the danger of falling or collapsing (e.g., Figure 15, showing a butterfly falling downwards), as well as sky motifs and a lack of grounding (see Figure 4).

Other expressions of falling or seepage were observed in artworks with incomplete frames: in artworks where part of the frame was missing underneath the figure (e.g., Figure 4) and in artworks featuring incompletely outlined faces that flowed downwards. The frame within the artwork motif can indicate a need for holding or an attempt to hold different parts of the work.

### 8.8. Experiencing Change and Expressions of Change

All the children described the studio as a place that helped them regulate their emotions; specifically, they recalled experiences in which they arrived upset or frustrated and were calmer when they left. This is how Dana talked about the possibility of regulating anger and aggression by working with appropriate materials: “Yes, if I’m nervous, for example the task “Come Out on Top”. I was upset with someone, and I just took a hammer and hammered the wood”.

In the interviews, some children described the process they underwent while creating the artwork. For example, Tom said: “At the beginning [of the year] I was just scribbling and fooling around but then [towards the end of the year] I made signs”.

An examination of the artworks over time revealed changes in regard to two subthemes: loneliness and the movement between primary to controlled modes of expression. In regard to the subtheme of loneliness, a progression was detected in the artworks of two participants, from depictions of a single object to artworks that contained relationships between several objects (e.g., Dana’s first and last artworks, Figure 2 and Figure 16). Additionally, in terms of holding, Dana’s first artwork had a full frame made of wood with glass, while her last work used a partial frame made from textile.

Regarding the movements between poles of control and primary expression modes, an examination of the artworks revealed that two participants demonstrated an increased ability to control the material in the course of the year; for example, Keren’s artworks demonstrated primary expression modes and a preference for liquid materials at the beginning and middle of the year, with better control and use of mediating tools (brushes, stencils, and writing) by the end of the year. It can be assumed that the children’s ability to move between disorganization and control in their artwork (i.e., expressing internal chaos and regress to early developmental needs on the one hand, and the need for control over internal and external realities on the other) demonstrated that work in the studio enabled access to different and opposing experiences. The very movement between these situations can also indicate development and growth, and reflect changes in emotional experiences over time.

## 9. Discussion

The children who participated in this study were all coping with the loss of their home and family and with relational trauma as a result of maltreatment and neglect. They were all struggling to adapt to the foster-care environment, and the open studio was one of the therapeutic settings that they chose for themselves. The main findings of the present study are based on the nonverbal expression mode. The artworks were rich in terms of materials, techniques, forms, sizes, colors, and symbolic images. The fact that these participants created 255 artworks may show that they used the open studio for an ongoing and continuous journey. The discussion discusses the themes from the aspect of coping with loss and trauma, and the evolvement of an alternative envelope through the artmaking processes in the open studio setting. It also deals with this study’s limitations and offers recommendations for further research.

### 9.1. Nonverbal Expressions of Trauma and Loss in the Open Studio

Trauma researchers agree that traumatic memories are primarily stored in the nonverbal part of the right hemisphere of the brain [15,52,53,54]. In addition, contemporary studies have found that talking about trauma can be a difficult experience at best; at worst, it reactivates the traumatic experience [55]. This is even more apparent when the trauma occurs within the child’s home and she/he is not protected by caregivers.

The present study showed that participants were preoccupied with the theme of close relationships, and that this theme manifested itself mainly through art expression. For these children, close relationships can mean both the loss of family members as well as traumatic experiences in which family members were involved. Multiple artworks representing close relationships may thus demonstrate the children’s efforts to work through memories of loss and trauma. Studies show that children in out-of-home placement reported a positive experience of their parents, despite significant reports of trauma as well as an intense longing for their biological family [56]. The intense longing for families of origin can be observed in artwork that expressed and emphasized the love that children have for family members (for example, artworks in which the word “mother” appears with a big heart).

In many foster-care services, the loss of parents and homes is frequently disenfranchised [57]; therefore, the grieving child should be helped in this respect. Therapeutically speaking, loss is an issue that should be addressed, according to the Two-Track Model of Bereavement [58]. Bereaved clients have two needs: to continue the relationship with their lost loved ones and to adapt to the new realities without them. From this clinical perspective, delegitimatizing the need to mourn and mentally preserve the relationship with the lost object may impair the process of adapting and functioning [58]. In a similar way, the Dual-Process Model (DPM) of coping with bereavement [59,60] asserts that a bereaved individual needs to deal simultaneously with loss-oriented and restoration-oriented stressors. We may thus assume that the children perceived the open studio as an enabling space for coping with and expressing the loss of family members in a nonverbal way. The research shows that art therapists perceive the art medium as a space for clients’ grief work (e.g., [61]).

In this study, the loss of family members was bound together with trauma connected to familial relationships. The experiences of relational trauma are particularly hard to assimilate. The findings of this study showed that in the artworks representing close relationships, various materials or techniques reflected negative affect, such as aggression and sadness through technique (e.g., hammering on nails), the materials (e.g., metal nails), and/or by integration of art’s content and form. These may reflect the children’s attempts to process this subject through nonverbal means. Art enables the embedded and material expression of intense longing alongside complex feelings of anger and frustration [62].

In addition, the children’s artworks reflected a profound experience of loneliness. Neglected children may heavily rely on themselves as a survival mechanism that helps them overcome the pain of loneliness [63]. Thus, the children’s ability to create and communicate the experience of loneliness may be thea first step in their healing process.

In summary, we may assume that the art medium was chosen by these children as a space in which they could express and communicate painful memories and experiences relating to relational trauma and loss. These painful themes were then observed and witnessed by the art therapists and other children who simultaneously worked in the open studio, and may have prompted verbal sharing. The research has shown evidence that the expression of negative emotions related to risky environments and trauma may have a positive impact by reducing the effects of the cumulative risk load (e.g., [64,65]).

### 9.2. The Formation of an Alternative Envelope in Which the Need to Be Held, to Be Seen, and Move between Opposite Poles Is Expressed

Since all the participants in the current study were removed from home due to maltreatment and/or neglect, we can assume that many of their central developmental needs were not met, and that they had to cope with stressors relating to the loss of their family envelope [10]. In addition, the traumatic experience itself disrupted the continuity and experience of life, as well as development [12,13]. The themes emerging in this study also indicate that the studio enabled the exploration of issues and dialectical moves. For example, through the creation of multiple frames featured in the artworks, which hold and contain the internal parts of the work, and through various containers that were used to contain liquid materials, the children were engaged in inserting something into a construct that was supposed to hold it, and were also engaged with the opposite pole—the collapsing of frames, the leaking of materials. These engagements may represent their need to be held, to belong, as opposed to being an outsider, being apart, or disintegrating, and the artistic enactments of their experiences as maltreated and neglected children. The studio and the children’s experiences with the materials may have functioned as a holding mechanism for the children, as a place where the client can gather the different parts of the self into one place [48]. According to Farrell-Kirk [51], the container in art therapy is also a metaphor for the space of understanding created in the therapeutic relationship between the therapist and client.

Cohn [63] describes parental neglect as the failure to mirror the child. In therapy, these clients’ experiences of being observed by the therapist may elicit negative emotions, such as embarrassment, shame, and humiliation [66]. The need to be noticed and to be observed that arises from the children’s artwork (through prominent shiny materials, and motifs of seeing) may indicate that needs for mirroring, for being seen and found, were not experienced in the past. One of the prominent roles of the art therapist in the open studio is to witness the participant’s art process and product [28]. In addition, displaying the artworks at the end-of-year exhibition adds a communal and social dimension to this process, and based on their comments in the interviews, the exhibition was probably a positive, corrective, and memorable experience for many of the children.

Contrary to the wish to be seen or found, various expressions of cover and concealment were present in the artworks. The participants used multiple layers to hide or bury objects within the materials, or covered initial drawings with multiple layers of paint. A clinical assumption may be that through burial and concealment, shameful parts and/or secrets or threatening parts of the self were symbolically kept safe [47].

In a recent study about the experience of adults who grew up in the shadow of a secret in their childhood, art-based representations included motifs of visibility, sight, and covering [67]. Children at-risk deal with familial secrets, as well as with events that they cannot fully grasp and understand. Indeed, relational trauma may leave the child with feelings of guilt and shame, and these feelings are particularly hard to articulate verbally [63]. Various acts of revealing and covering may be opportunities for the creator to explore this continuum. Similar to the movement between poles of visibility and holding, there was also movement between art expressions that conveyed control (e.g., a symmetrical graphic image) versus primary modes of expressions (e.g., smearing).

In terms of trauma, the prominent kinesthetic sensory dimension identified in the children’s works indicated the promotion and restoration of movement, which is the opposite of the freezing process; in trauma, kinesthetic sensory work can release pressure, awaken the senses, cause embedded pre-verbal memories to resurface, and create a rhythm of healing [68].

At the extreme pole of primary modes of expression, the artworks expressed chaos, which may represent the children’s emotional experiences of disorder in their life; primary developmental needs that were nor met; a sense of vagueness; mixed feelings; as well as the hope that in the open studio they might be welcomed and acknowledged. At the opposite pole, the findings show that the artworks reflect the need for control through motifs of order, repetition, and symmetry. This may express an attempt to bring order to a life experience of trauma and chaos, to control reality (at least in the drawing), and create a shield against helplessness.

When the children fail to develop a sense of control and stability, they become helpless [54]. This study found motifs of “superpowers” in some of the children’s artworks, represented by crowns or superheroes. This may reflect the children’s preoccupation with planning, order, and control, elements that were lacking in their families of origin.

### 9.3. Experiences of Change in the Open Studio

Experiences of change came up in the participants’ interviews when the children recalled entering the studio whilst feeling upset or angry, and leaving feeling calmer and/or content. This documents an emotion-regulation process that may be associated with the open studio setting [69] or/and the process of art making that is known as enabling fun for children in foster care [20,21], as well as a reduction in arousal and stress among a normative sample (e.g., [69]). Deficits in emotional regulation, as well as negative emotionality, are characteristic of maltreated and neglected children; thus, therapy must address these issues [70]. These repeated experiences of affect regulation in the open studio may demonstrate a new emerging ability among these children.

The artwork of all the study’s participants revealed movement between different levels of expression according to the ETC model, which can indicate change and transformation [71]. Art therapists report the transition between expressive levels in therapy as part of the processes of transformation and change in their clients [68,72,73]. To summarize, the ability to move between chaotic primary expressions in art to more controlled perceptual expressions, and vice versa, may contribute to the enhancement of emotional self-efficacy as an important aspect of art therapy treatment for traumatized individuals [74].

An examination of the changes in the children’s artworks over time revealed that, for two participants, there was a shift from works that expressed loneliness to works that contained relationships; however, these images were floating, without a background or ground. A study conducted in Mexico on children who were removed from the home and placed in boarding schools observed that the home was described as “almost home” and the staff and other children as “almost family” [75]. This can explain the representations of relationships in the artwork used in this study, which featured hovering figures, as reflecting the ability to “almost” be in a relationship, or in the process of establishing new relationships.

### 9.4. Limitations of This Study and Recommendations for Future Research

The main limitation of this qualitative study was its small sample size. At the same time, the significant and representative number of artworks (82) that were analyzed and chosen out of the 255 artworks strengthened the validity of the themes, which were refined through the thorough analysis conducted in this study. In addition, the study mainly relied on nonverbal expressions of the artworks; thus, subjective biases might have influenced the final analysis. To counter this bias, the researchers cooperated in the process of analyzing the data to ensure intersubjective validity [76].

Another limitation was the scarce interview content: the children’s age or lack of mentalization skills made it difficult for them to expand on their experiences in the studio. To some extent, the richness of the artwork compensated for this lack, but increased our reliance on analysis and interpretation. In view of the limitations, we recommend conducting mixed-methods studies with larger sample sizes, and the integration of valid measures to assess the children’s emotional and behavioral functions. We also recommend examining studio activities from the perspective of the therapists who can provide their impressions of their interactions with the children.

### 9.5. Practical Implications for the Research and Clinical Work

The many artworks created by the participants throughout the year present a profound and complex picture of their subjective experience in the studio and the issues that preoccupied them. The present study reinforced the importance of nonverbal expression as a research tool, especially with at-risk children who experienced trauma, and for whom this trauma was less accessible through verbal channels.

From a clinical perspective, the present study demonstrated that the open studio may serve as a meaningful therapeutic framework for at-risk children in foster care. The richness of their artwork may reflect a good fit between participant needs and the open-studio setting. Through visual communication, children could have learned affect-regulation and gained access to their inner worlds, their own voices, and to vital partnerships with the other creators in that space, not to mention with the art therapists who held this space, witnessed the evolution of these artworks, and finally created the art exhibition, with the children as joint curators.

## Figures and Tables

**Figure 1 children-09-01218-f001:**
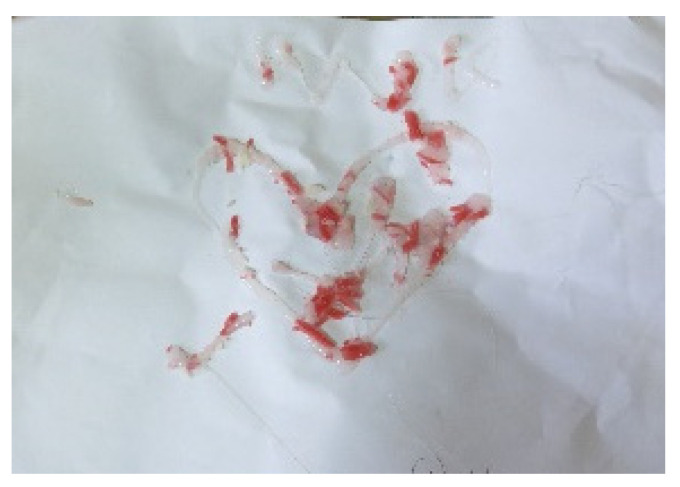
Keren and Tamar. Warm glue and red material, 21/29.7 cm.

**Figure 2 children-09-01218-f002:**
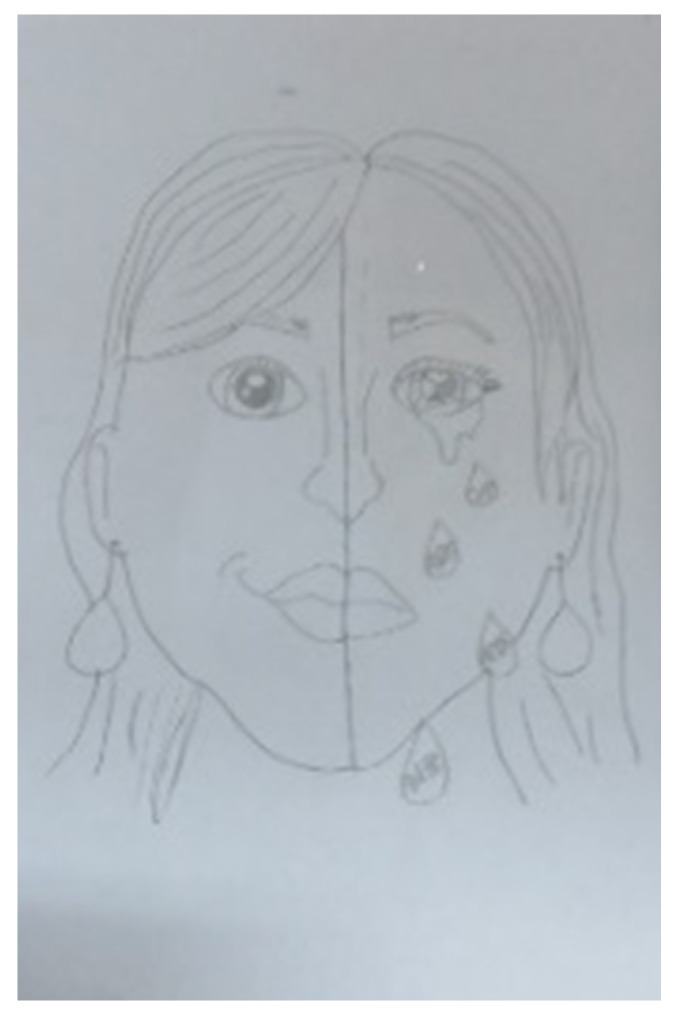
Dana. Pencil, 21/29.7 cm.

**Figure 3 children-09-01218-f003:**
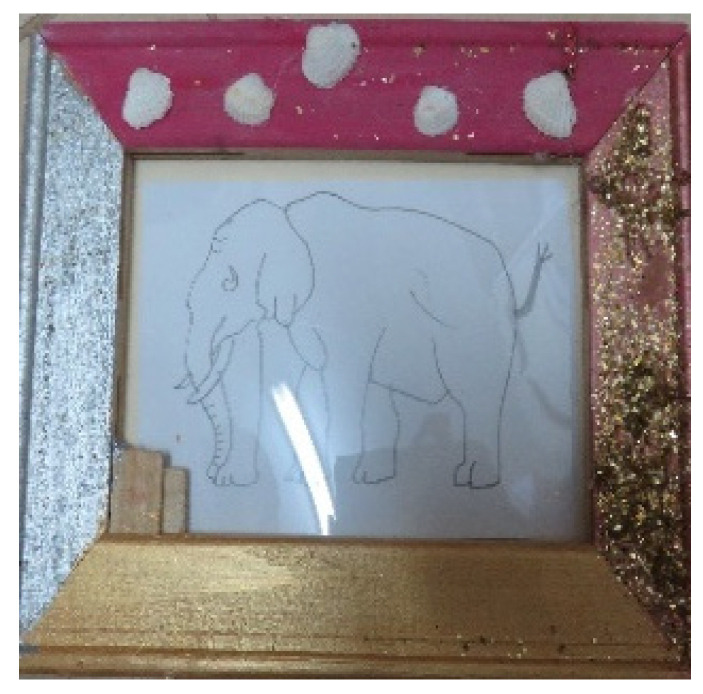
Dana. Paper, pencil, wooden frames, gouache paints, sequins, shells, and glue, 15/20 cm.

**Figure 4 children-09-01218-f004:**
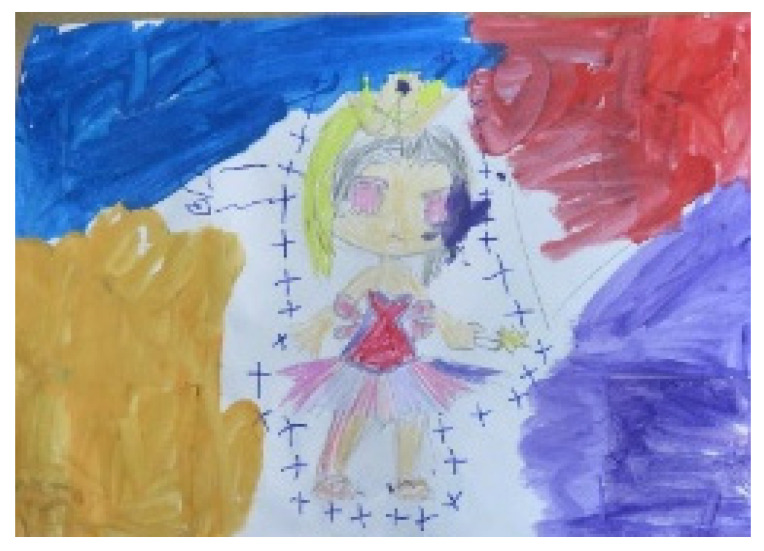
Tamar. Gouache pencils, markers, and 70/50 cm sheet of paper.

**Figure 5 children-09-01218-f005:**
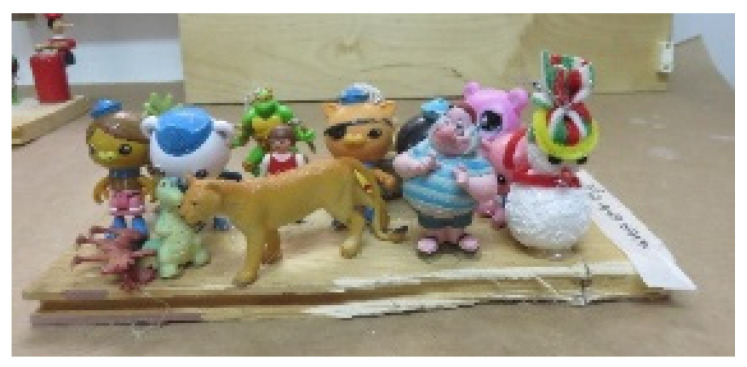
Alex. Readymade figures on a wooden surface, 20/40 cm.

**Figure 6 children-09-01218-f006:**
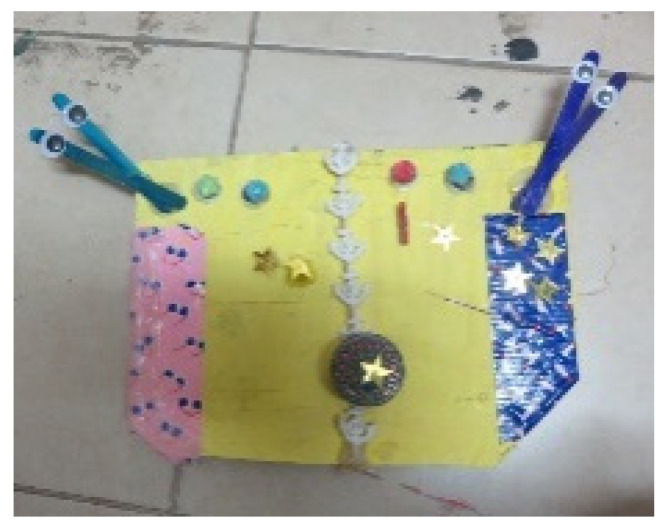
Dana and Tom. Readymade materials, wallpaper, and gouache, 40/50 cm.

**Figure 7 children-09-01218-f007:**
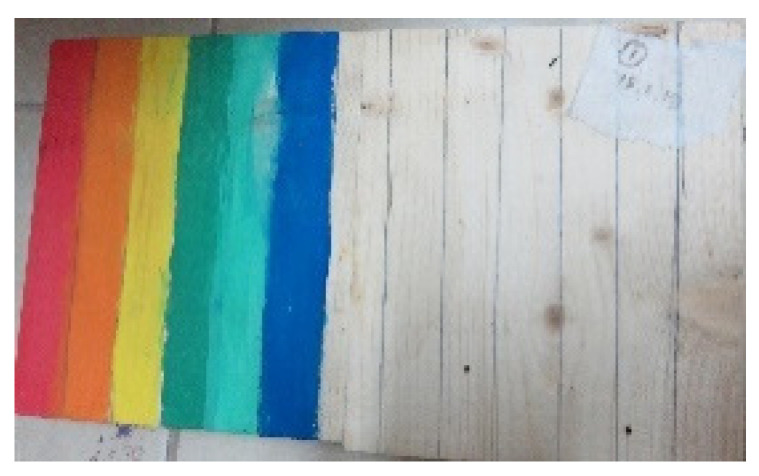
Tom. Wood, gouache, and pencil, 30/60 cm.

**Figure 8 children-09-01218-f008:**
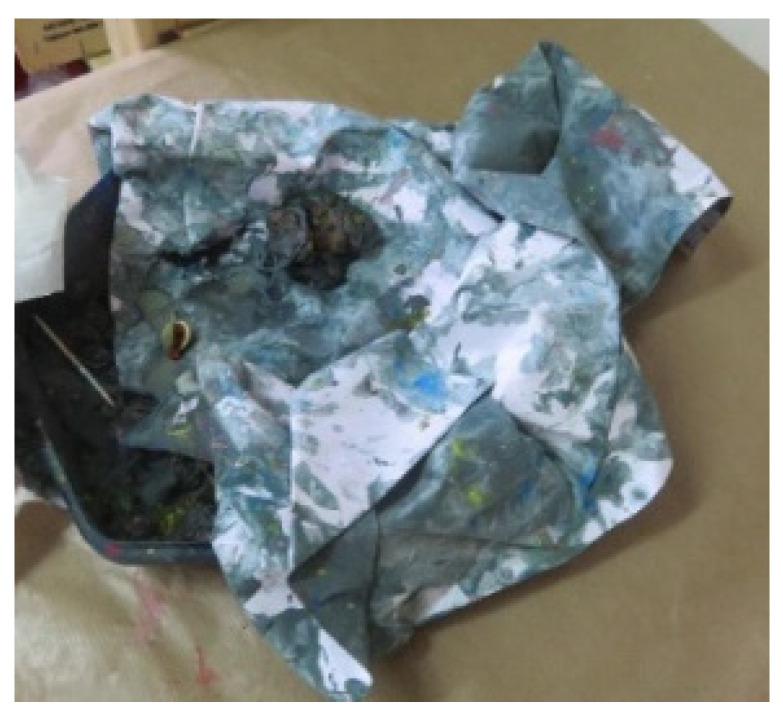
Keren. Plastic container, paper, and gouache, 30/40 cm.

**Figure 9 children-09-01218-f009:**
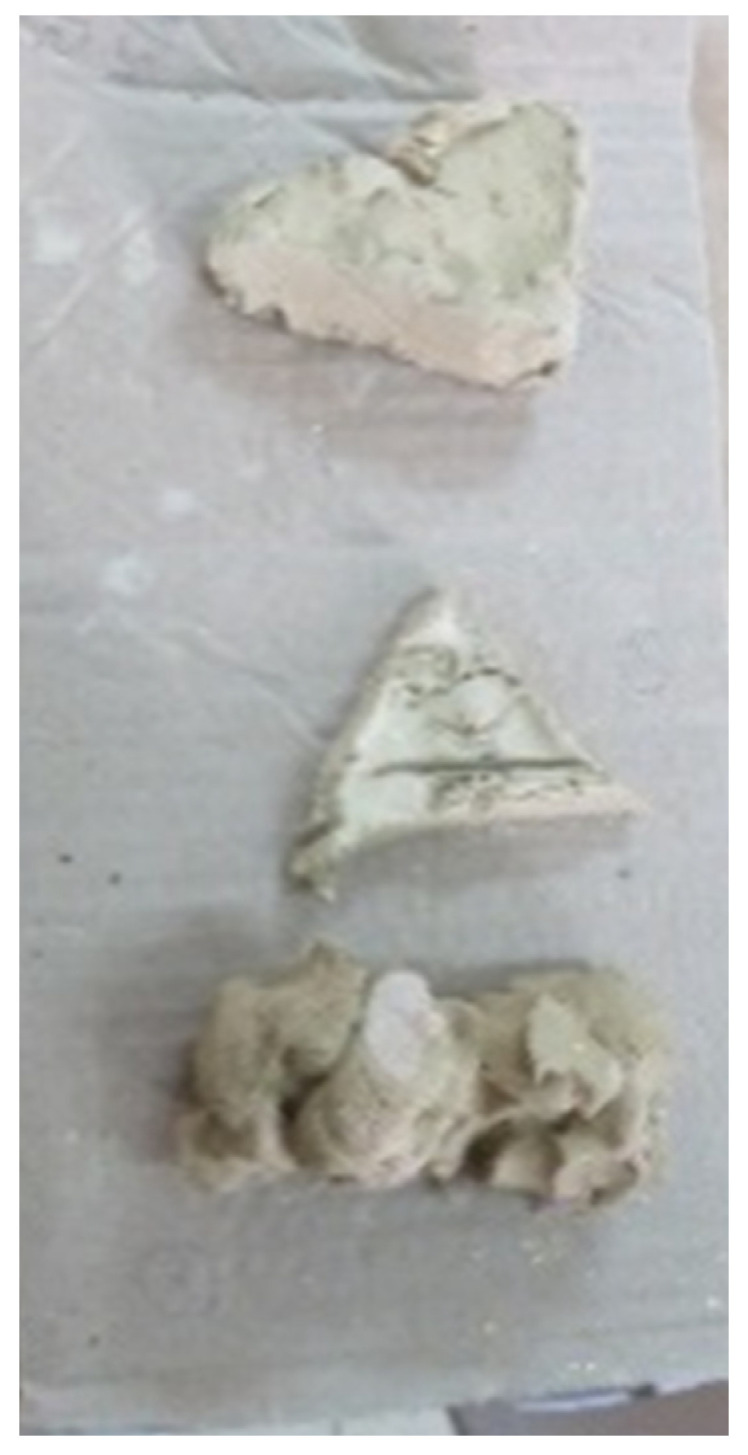
Tamar. A series of three objects, clay.

**Figure 10 children-09-01218-f010:**
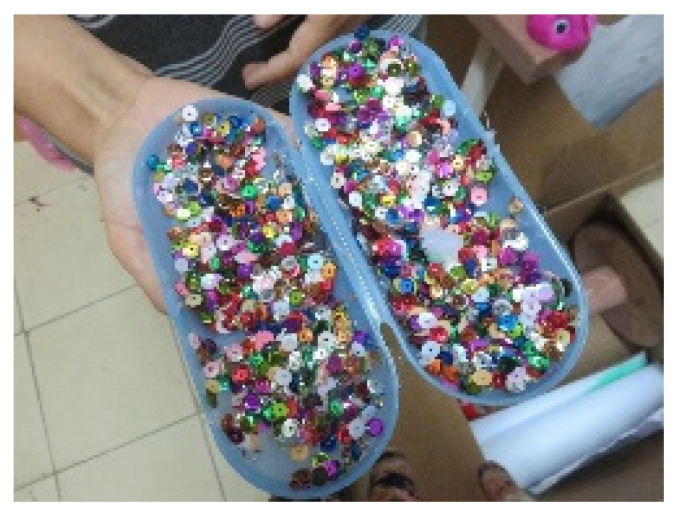
Tom. Pencil case, sequins, and glue, 15/25 cm.

**Figure 11 children-09-01218-f011:**
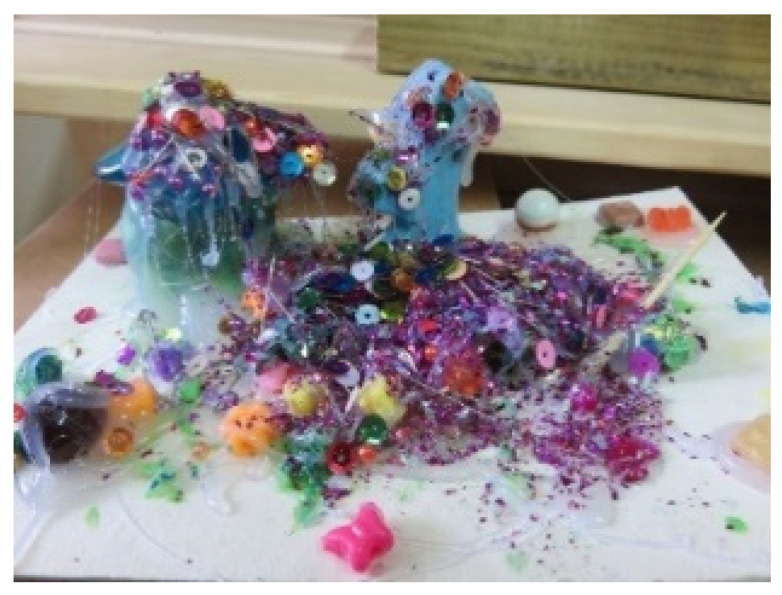
Keren. Ready-made materials, sequins, and beads on corrugated cardboard.

**Figure 12 children-09-01218-f012:**
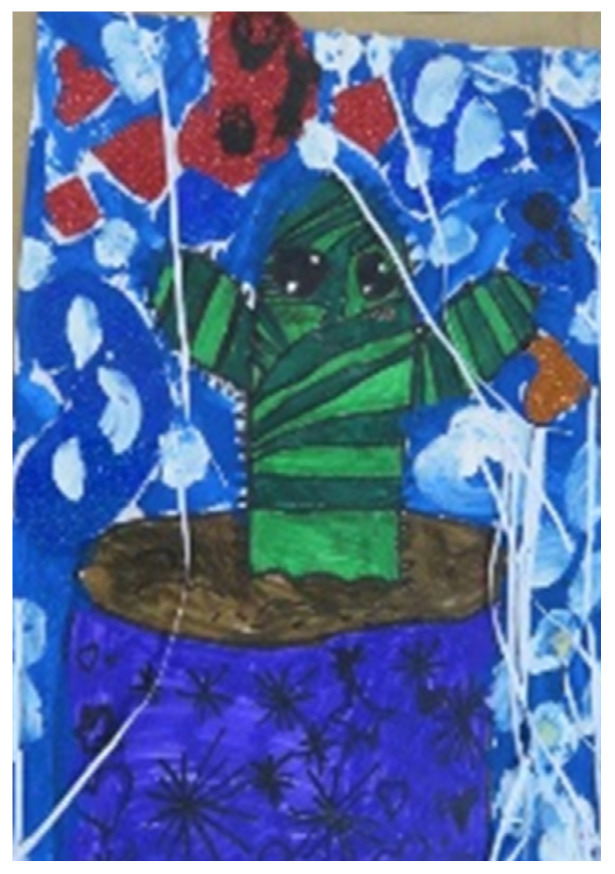
Tamar. Gouache, markers, paper, and ¼ sheet of paper.

**Figure 13 children-09-01218-f013:**
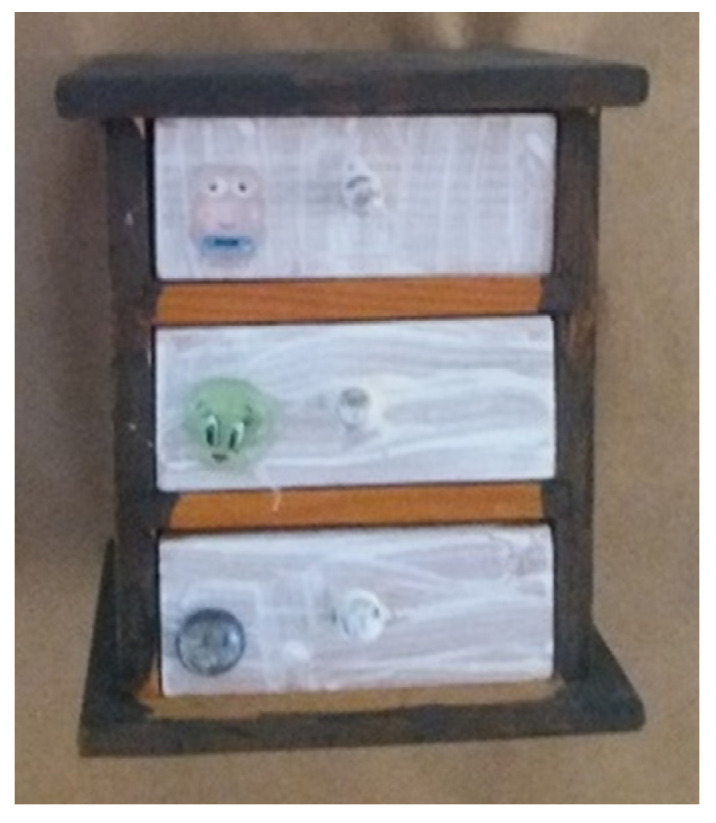
Dana. Readymade drawers, gouache, and beads, 15 × 25 × 5 cm.

**Figure 14 children-09-01218-f014:**
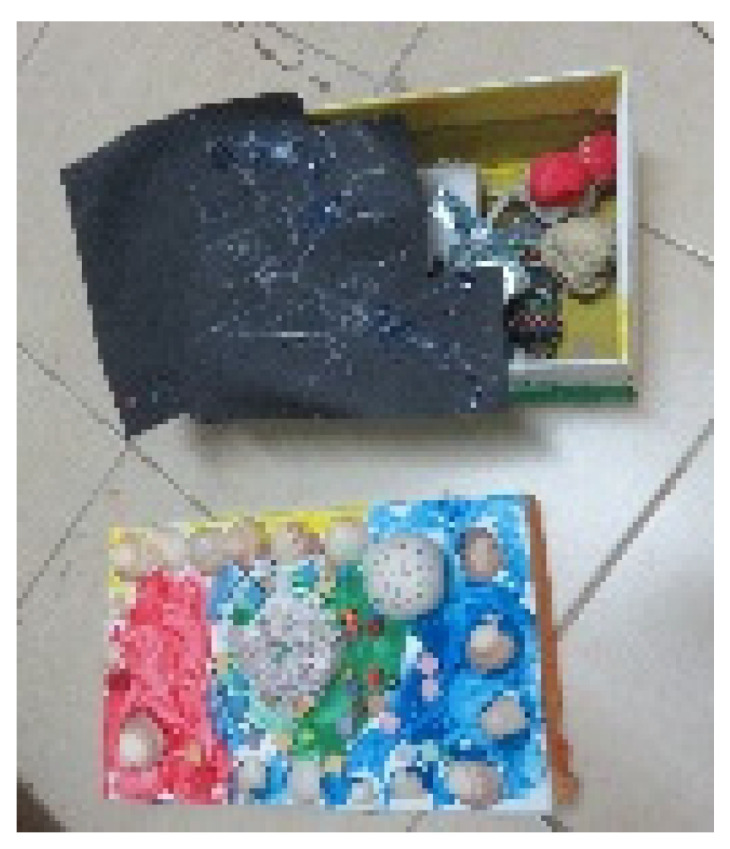
Alex. Box of gouache, readymade materials, shells, cardboard, glue, and clay.

**Figure 15 children-09-01218-f015:**
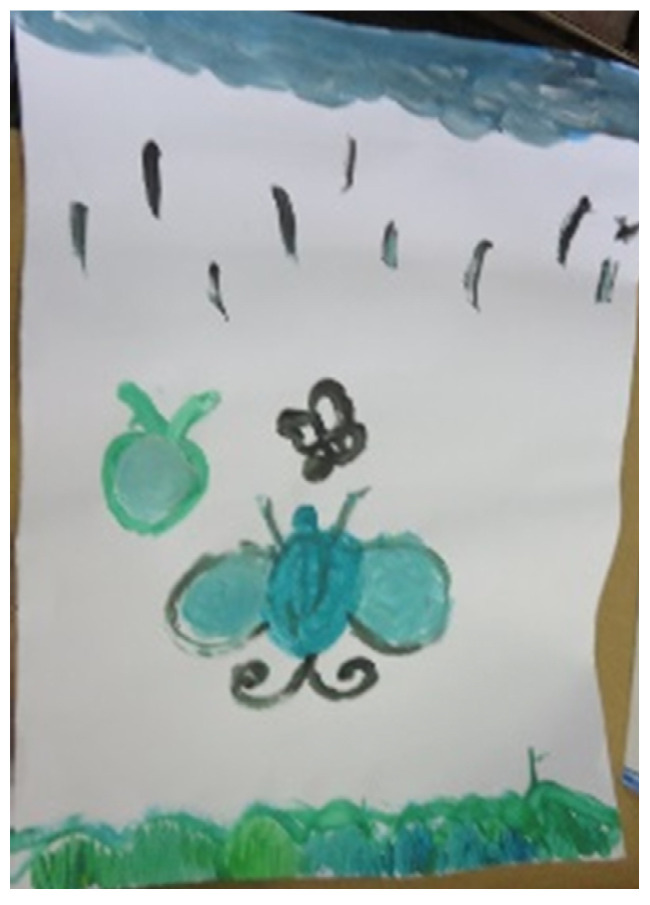
Dana. Gouache and ½ sheet of paper.

**Figure 16 children-09-01218-f016:**
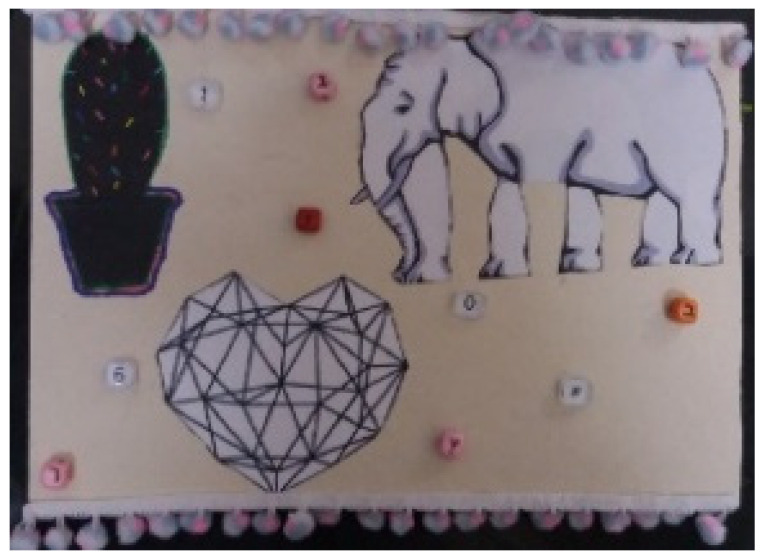
Dana. Corrugated cardboard, bits of paper, markers, and ready-made materials, 21/29.7 cm.

**Table 1 children-09-01218-t001:** Participants’ age, gender, attendance, and number of artworks produced.

Pseudonym	Age	Gender	Number of Studio Sessions Attended	Number of Artworks
Dana	10	Girl	42	47
Tom	9	Boy	21	21
Keren	8	Girl	31	44
Tamar	7	Girl	41	109
Alex	8	Girl	21	44

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
