# Peer review of "Subjective Experiences of At-Risk Children Living in a Foster-Care Village Who Participated in an Open Studio"

_children, 2022, doi:10.3390/children9081218_

Round 1

Reviewer 1 Report

The introduction can benefit from statistics regarding children at risk and context regarding placement (70% of what?; line 36) and line 39 (Studies have found a high incidence of psychological, social and educational problems...). Line 46 needs elaboration (containing aspect of the family for the child). Please provide examples. Line 82 missing citation (Finkel & , 2020) and line 86, 88, 103 (Finkel & XXX, 2020), Line 98 (Block, et al., 2005). Line 576 (XXX 2015). Line 579 (Collie, et al., 2006).

Question about the raters. Were they art therapist and trained in the rating system (Line 132-134). Clarify the qualifications and training of the raters. What was the rater reliability? Who is "we"?

The presentation of Findings is a bit confusing. Maybe revisit this section. Artwork should be presented in order they are mentioned intext.

Overall its a good study that needs minor changes and reorganization of Findings.

Author Response

We wish to thank deeply the three reviewers of our manuscript. We feel that their comments, questions, and suggestions, contributed to our efforts to write a better manuscript.

Revisions (in red) made following the suggestions of Reviewer #1

  • (70% of what?; line 36)- thank you for that, we added this:

“In Israel, 70.1% of out-of-home placements involve boarding schools…”

  • The introduction can benefit from statistics regarding children at risk and context regarding placement and line 39 (Studies have found a high incidence of psychological, social and educational problems...). we have added some statistics as requested:

“Studies have found a high incidence of psychological, social and educational problems among children in out-of-home placement in comparison to the general population, for example a recent systematic review and meta-analysis of prevalence found that suicide attempts are more than three times as likely in children and young adults that were in out-of-home care [5]. Health screening of 122 young children in out-of-home care found that 54% had behavioral/emotional problems (e.g., poor self-esteem, anxiety, grief, problematic interpersonal relationships, depression), 33% had a speech delay, the hearing test of 30% was abnormal, and 28% failed their vision test [6]. Recent studies also underscore the higher prevalence of sleep problems as related also to mental health problems and crime among out-of-home care children [7] and young people who were in out-of-home care as children [8]  

  • Line 46 needs elaboration (containing aspect of the family for the child). Please provide examples. We agree that it was vague, so we add further explanations:

Houzel [10] coined the term “family envelope”, which provides the experiences of belonging and connectedness, but also the permission to differentiate themselves and to form individual identities.  He suggested that in cases of dysfunctional families, therapeutic services may provide a “widened envelope,” by joint efforts of therapeutic professionals who provide a sense of belongingness together with the permission for children to differentiate themselves [10].

  • Line 82 missing citation (Finkel & , 2020) and line 86, 88, 103 (Finkel & XXX, 2020), Line 98 (Block, et al., 2005). Line 576 (XXX 2015). Line 579 (Collie, et al., 2006). – We have added the missing name (unmasked the first author). We received the journal request to number all the references, so these are all completed now in the end of the manuscript.

  • Question about the raters. Were they art therapist and trained in the rating system (Line 132-134). Clarify the qualifications and training of the raters. What was the rater reliability? Who is "we"? - We wish to thank you for this, it was also one of the comments of the second reviewer. We have added more information regarding the process of rating the artworks.

Two art therapy students (the second author was one of them) and an experienced art therapist and researcher (the first author) practiced the ratings of the three research tools for this study, and for an additional study with adolescents [35]. After the raters had reached an agreement in regard to the phenomenological aspects of the artworks during analysis, via careful observation, analysis, discussions, and continued integration of the relevant theoretical models, the second author continued the rating independently, and the second author participated in the reduction process of the integration into themes. The data-codes were inserted into excel table for further data analysis and integration.

  • The presentation of Findings is a bit confusing. Maybe revisit this section. Artwork should be presented in order they are mentioned intext. We agree, and revisited this, and arranged the artworks by their order as they are presented in the text.

Reviewer 2 Report

Dear authors, first of all, thank you very much for allowing me to review the article and I hope that the contributions I refer to will be of interest to you and will help to improve it.

Regarding the theoretical introduction, it should be adjusted to the format of the journal. In general the theoretical framework is obsolete but the one referring to art therapy in particular is deficient and needs updating of references and scientific articles of impact journals. Art therapy is a field that has grown a lot in recent years and we can find numerous studies and references so it is necessary to update the article, as well as the inclusion of references in the rest of the epigraphs more updated.

The method section is clearly explained, however, it is observed that the sample is excessively small, as already mentioned in the limitations of the study, the instrument section would be important to explain which variables are measured and if these are some kind of factor.

With respect to the presentation of results, it would be interesting to homogenize common responses of the variables through the use of percentages or graphs that would allow us to draw immediate conclusions from the results obtained. I believe that it is necessary in the procedure to describe very clearly the analysis because and to comment on how the researcher has evaluated the results, since at times it is confusing and may make the reader think that it is an article very dependent on the researcher's bias.

Author Response

We wish to thank deeply the three reviewers of our manuscript. We feel that their comments, questions, and suggestions, contributed to our efforts to write a better manuscript.

Regarding the theoretical introduction, it should be adjusted to the format of the journal. In general the theoretical framework is obsolete but the one referring to art therapy in particular is deficient and needs updating of references and scientific articles of impact journals. Art therapy is a field that has grown a lot in recent years and we can find numerous studies and references so it is necessary to update the article, as well as the inclusion of references in the rest of the epigraphs more updated. We agree that the section about art therapy in the introduction had deficient, and we revised it to include up to date findings. We omitted the old references that were taken from book-chapters. Please see the changes in red.

The visual art medium as a therapeutic tool for treating children at risk

Although expressive arts are a developmentally appropriate activity for children that  enable their voice to be heard (Desmond et al., 2015), and a high percentage of arts therapists work with children (e.g., Moula et al., 2022), the art therapy research for traumatized children constitutes mainly in the form of case studies and qualitative research (see the review of Van Westrhenen & Fritz, 2014). Creative expression for traumatized children has been found to be an effective therapeutic tool (Coholic et al., 2009a, 2009b), because children communicate their emotions and thoughts through non-verbal means (DiSunno et al., 2011; Desmond, et al., 2015; Hecker et al., 2010); moreover, through the art expression,children gain more knowledge and awareness of themselves, and develop their self-esteem as well as socialization skills (Coholic et al., 2009a). Resiliency refers to the ability to adapt despite experiencing deficiency and severe distress (Luthar, et al., 2000). Art therapy may promote resiliency among clients who suffer from adverse life events (Disunno et al., 2011; Worrall & Jerry, 2007). Artwork therefore may allow clients to play an active role in their own personal therapeutic processes (Steele, 2009).

The method section is clearly explained, however, it is observed that the sample is excessively small, as already mentioned in the limitations of the study, the instrument section would be important to explain which variables are measured and if these are some kind of factor. We agree with this comment, and revised the method section to include additional information:

Observation Rating Scales Sheet for Art Therapy Practice: Sections B1 and B2: Observation and analysis of drawings and three-dimensional artworks—sculpture/structure/textile [41].

This tool is used for art therapy education and practice; the rater is required to answer yes/no to a detailed list of formal and content features. The tool contains five main formal aspects: overall impression (for example- artwork with repetitions); line quality (for example, thick or weak line, the presence of eraser); color qualities (for example, separated versus mixed colors); space (for example, the amount of space used, and the presence of a frame); and finally, forms (for example, the presence of geometrical forms). In terms of content, the rater is asked to rate the artwork for its realism/abstraction and symbolism.

For this tool Analysis according to the Expressive Therapies Continuum model ETC- The Expressive Therapies Continuum [42,43]. We add this:

The first and basic level ranges from kinesthetic components (for instance, evidence of smearing, or pressing on the material) to sensory components (for instance, evidence of direct touches on the material, such as finger marks on clay); the second level ranges from perceptual components (for instance, evidence of differentiated forms and colors, contour-lines etc.) to affective components (for instance, evidence of the expressive use of colors); the third level ranges from cognitive components (for instance, the addition of written words) to symbolic components (for instance, the expression of a metaphor, the combination of realism and abstraction, and more);

With respect to the presentation of results, it would be interesting to homogenize common responses of the variables through the use of percentages or graphs that would allow us to draw immediate conclusions from the results obtained. We agree, and added the graph bar that was created by the excel table, that summarize all the study’s codes.

Bar Chart 1 illustrates the evidence of each of the four themes as they were detected in each child’s verbal and artwork expressions.

 I believe that it is necessary in the procedure to describe very clearly the analysis because and to comment on how the researcher has evaluated the results, since at times it is confusing and may make the reader think that it is an article very dependent on the researcher's bias. We agree with this comment, and this comment was also given by the first reviewer. We added more information about the process of analysis:

Two art therapy students (the second author was one of them) and an experienced art therapist and researcher (the first author) practiced the ratings of the three research tools for this study, and for an additional study with adolescents [35]. After the raters had reached an agreement in regard to the phenomenological aspects of the artworks during analysis, via careful observation, analysis, discussions, and continued integration of the relevant theoretical models, the second author continued the rating independently, and the second author participated in the reduction process of the integration into themes. The data-codes were inserted into excel table for further data analysis and integration.

We also added a more elaboration of the identification of change over time

Finally, for the purpose of examining the artworks for changes over time, the authors looked at the summary table and the individual summary of each child artworks, and identified expressions of changes over time, including explicit verbal expressions in the children’s interviews, and visual expressions in their artworks that showed a change in formal art elements and/or content representations. 

Reviewer 3 Report

I have no comment on the article. It is written clearly and comprehensibly. The topic is interesting.

The main positive can be considered that the theoretical facts are suitably supported by practical examples.

The article will be interesting for readers.

Author Response

R3

I have no comment on the article. It is written clearly and comprehensibly. The topic is interesting.

The main positive can be considered that the theoretical facts are suitably supported by practical examples.

The article will be interesting for readers.

Thank you so much for this positive comment, we revised our manuscript according the comments of the two reviewers, and hope our manuscript is now more elaborated, clear, and contributing.